# Adjuvant Effect of Whole-Cell Pertussis Component on Tetanus Toxoid Potency in Murine Model

**DOI:** 10.3390/biomedicines11071795

**Published:** 2023-06-23

**Authors:** Marta Prygiel, Ewa Mosiej, Karol Wdowiak, Daniel Rabczenko, Aleksandra Anna Zasada

**Affiliations:** 1Department of Sera and Vaccines Evaluation, National Institute of Public Health NIH—National Research Institute, Chocimska 24, 00-791 Warsaw, Poland; emosiej@pzh.gov.pl (E.M.); kwdowiak@pzh.gov.pl (K.W.); azasada@pzh.gov.pl (A.A.Z.); 2Department-Center for Monitoring and Analyses of Population Health Status, National Institute of Public Health NIH—National Research Institute, Chocimska 24, 00-791 Warsaw, Poland; drabczenko@pzh.gov.pl

**Keywords:** adjuvant, vaccines, diphtheria, tetanus, pertussis, control laboratory

## Abstract

There is currently an increasing interest in the development of new-generation purified antigen-based vaccines with a higher safety profile compared to conventional inactivated vaccines. The main problem of subunit vaccines is their lower immunogenicity compared to whole-cell vaccines and inducing weaker and shorter-lasting immune responses. In this paper, the results of the assay of the potency of the tetanus component combined with the diphtheria component and whole-cell pertussis vaccine (DTwP), diphtheria and tetanus vaccine (DT), and in monovalent tetanus vaccine (T) are presented. In the mice model, an adjuvant impact of the whole-cell pertussis component on the immune response against tetanus was observed. It was noticed that the potency of tetanus component in the DTwP vaccine was significantly higher than tetanus potency in DT and T vaccines, despite the same bounding ability unit of the tetanus toxoid in the vaccine formulations. The levels of induction of tetanus antibodies by the tested vaccines were also examined. There were no differences in the induction of humoral responses against tetanus by tested vaccines. This publication discusses the possible mechanisms of impact of the whole-cell pertussis component on the other vaccine antigens and the positive and negative aspects of using the whole-cell pertussis component as an adjuvant.

## 1. Introduction

The Diphtheria–Tetanus–Pertussis (DTP) vaccine is one of the oldest human vaccines. It protects from three serious diseases—diphtheria, tetanus, and pertussis. Before the DTP vaccine was registered in 1948, monovalent vaccines containing each component separately had been used. The pertussis vaccine was licensed in 1914. Diphtheria toxoid was registered in 1926, while adsorbed tetanus toxoid was registered in 1937 [1]. It is estimated that, in 1995, DTP vaccines were produced by 63 manufacturers in 46 countries globally [2]. In 2006, over 500 million doses of DTP vaccines were manufactured worldwide [3].

Inactivated vaccines, including non-replicating T, D, and Per antigens, require an adjuvant to maximize the effectiveness and duration of antigen-specific immune responses. The vaccines used in this study contain aluminum hydroxide (alum) as an adjuvant. Alum has been used as an adjuvant since the 1920s when Glenny et al. observed that animals vaccinated with diphtheria toxoid precipitated with potassium alum were better protected against intradermal injections with diphtheria toxin than animals injected with diphtheria toxoid without alum. Since then, alum has become the most popular adjuvant licensed in many human vaccines [4]. One of the first trials of DTP vaccines showed that the wP vaccine is also a potent adjuvant, and combining the three antigens in DTwP improved the immunogenicity of the toxoids compared with separate administration [5,6].

Currently, Poland is the only EU and EEA country where the wP vaccine is used for primary immunization [7], and the DTwP vaccine is still submitted for official control batch release. The potency assay of the tetanus component is one of the tests required to be performed on the tetanus toxoid vaccine for batch release. This assay reflects the actual relative strength of a particular vaccine batch and is correlated with immunogenicity, clinical efficacy, and safety.

In this study, the results of a potency assay for 40 batches of tetanus vaccines, monovalent and combined with diphtheria and pertussis components, are shown. The humoral responses against tetanus in mice immunized with T, DT, and DTP vaccines have also been investigated.

## 2. Materials and Methods

### 2.1. Challenge Test in Mice

The tetanus component potency test was performed following the requirements of the monograph 2.7.8 “Assay of tetanus vaccine” of European Pharmacopoeia, method B “Challenge test in mice” [8]. The potency of tetanus is determined by comparing the dose of the tested vaccine required to protect mice from the effects of a subcutaneous injection of a paralyzing dose of tetanus toxin with the dose of a reference preparation, calibrated in International Units, needed to give the same protection. The reference vaccine has a known potency expressed in international units (IU). The potency of the reference vaccine has been tested against an international standard.

International Units (IU) is the activity contained in a specified amount of the International Standard for tetanus toxoid. The equivalence in International Units of the International Standard is provided by the World Health Organization (WHO).

#### 2.1.1. Vaccines Used for Immunization

Eleven batches of the diphtheria–tetanus-whole-cell–pertussis (DTwP) vaccine (trade name DTP, manufactured by IBSS Biomed S.A., Cracow, Poland), seven batches of the diphtheria–tetanus (DT) vaccine (trade name DT, manufactured by IBSS Biomed S.A., Cracow, Poland), and twenty-two batches of the tetanus (T) vaccine (trade name Tetana, manufactured by IBSS Biomed S.A., Cracow, Poland) were used in the study. The composition of the vaccines is shown in Table 1. As a reference, the AnaTe9 vaccine manufactured by IBSS Biomed S.A., Cracow, Poland, was applied.

A single human dose of each tested vaccine contains the same amount of tetanus toxoid (10 Lf) adsorbed on hydrated aluminum hydroxide. Lf means bounding ability unit, and it is the quantity of toxoid that flocculates one unit of the reference serum within a specified time.

#### 2.1.2. Immunization of Animals

Animal experiments were approved by the Polish Ethics Committee for Animal Experiments.

Five-week-old Swiss female mice from the same stock (breeding laboratory mice of Jacek Kołacz) were used. The animals were allocated to eight groups—four groups for the tested vaccine and four groups for the reference vaccine. Each group contained 10 animals. The mice were put into cages according to the randomization list. 

Four serial two-fold dilutions (8 IU, 4 IU, 2 IU, and 1 IU) of the reference material and the tested vaccine were performed using sterile sodium chloride solution 9 g/L. Each group of mice was vaccinated subcutaneously with 0.5 mL of each dilution of the reference vaccine and the tested vaccines. The immunization period was 28 days.

#### 2.1.3. Intoxication of Animals

After 28 days, the animals were challenged subcutaneously with 0.5 mL of the challenge tetanus toxin solution (IBSS Biomed S.A., Cracow, Poland) containing approximately 50 times the 50 percent paralytic dose in 0.5 mL (50 PD50). The tetanus toxin solution was diluted in a sterile peptone-buffered saline solution with pH 7.4. PD50 (50 percent paralytic dose)—the amount of toxin that, administered to a group of non-vaccinated mice, causes paralysis in 50% of them.

#### 2.1.4. Reading and Interpretation of Results

The animals were examined twice daily. All the mice showing definite symptoms of tetanus paralysis were removed and euthanized. The number of non-paralyzed mice four days after the intoxication was recorded, and potency was calculated. The potency of the vaccine was calculated based on the ratio of challenged non-paralyzed animals in each group of the tested mice in relation to the activity of the reference vaccine. The activity of the tested vaccine relative to the potency of the reference vaccine using a parallel-line quantitative analysis was calculated with the CombiStats Software (version 7.0, European Directorate for the Quality of Medicines & HealthCare, Strasbourg, France). The potency of the tested vaccine was expressed in International Units as the estimated potency and the lower and higher confidence limits (*p* = 0.95). 

The acceptance criteria for the potency of the tetanus vaccine, according to the European Pharmacopeia, are: the lower confidence limit (*p* = 0.95) of the estimated potency of the vaccine is not less than 40 IU (International Units) per single human dose. 

#### 2.1.5. Requirements for a Valid Assay

The potency assay was regarded as valid when the following requirements were met:

(i) For both the study vaccine and the reference vaccine, the 50 percent effective dose (ED50) lies between the largest and lowest doses of the preparations administered to the animals; the ED50 (effective dose) is defined as the amount of vaccine that protects 50% of the animals from the effects of tetanus toxin. (ii) The confidence limits (*p* = 0.95) of the estimated potency are not less than 50 percent and not more than 200 percent of the estimated potency. (iii) The statistical analysis shows a significant slope and no variation from parallelism and linearity of the dose-response curves.

### 2.2. Assessment of Anti-TT (Tetanus Toxoid) IgG Levels in Mouse Sera

#### 2.2.1. Immunization of Animals

Five-week-old Swiss female mice were allocated to four groups—three groups for each tested vaccine (described above T, DT, and DTP vaccines) and a control group. Each group contained 10 mice. The treatment groups were immunized subcutaneously with 0.5 mL of vaccine dose containing 8 IU (vaccine dilution ensures protection of 100% vaccinated animals against the administration of the challenge tetanus toxin solution in the challenge test). The control group was injected with the sterile sodium chloride solution 9 g/L used to dilute the tested vaccines.

#### 2.2.2. Blood Collection 

Blood sampling was performed 28 days after immunization. Blood was collected from anesthetized mice (a mixture of ketamine −0.8 mg/mouse and xylazine −0.1 mg/mouse) with the plexus of the ocular vessels. After the collection, the blood was incubated at 37 ± 1 °C for 3 h and then centrifuged for 10 min at 1200× *g*. The mice serum was stored at −20 °C until the ELISA was conducted. At the end of the experiment, animals were euthanized by carbon dioxide overdosing.

#### 2.2.3. Determination of Anti-TT IgG in Mouse Sera by ELISA Assay

The antibody levels against tetanus antigen were determined by an in-house validated ELISA method. Maxisorp 96-well plates (NUNC, Roskilde, Denmark) were coated with 1 ng/µL reference material of tetanus toxoid (NIBSC, UK) dissolved in a 0.05 M carbonate buffer (Sigma–Aldrich, St. Louis, MO, USA). The antigen coating of the plate was performed by incubation overnight in a humid chamber at 5 ± 3 °C. Subsequently, non-specific binding sites were blocked with a solution of 5% nonfat milk (AppliChem, Ottoweg, Germany). Tetanus toxin monoclonal antibody (Thermo Fisher Scientific, Roseville, CA, USA) was used to generate the standard antibody curve. Two-fold standard dilutions in the range from 1/4000 to 1/128,000 were used. The negative assay control (serum from the non-immunized mice) was used to control the correctness of the assay performance. The plate was incubated for 90 min at 37 ± 1 °C. After washing, the 1/2000 dilution of goat anti-mouse secondary IgG peroxidase conjugate antibody (Sigma–Aldrich, USA) was added. The conjugate was incubated for 60 min at 37 ± 1 °C. Next, an OPD substrate solution (Sigma–Aldrich, USA) was added. The plate was incubated for 30 min in the dark at room temperature. The assay was stopped by adding 1 N sulfuric acid (DiaSorin, Saluggia, Italy). Absorbance was read at a 490 nm wavelength using an ELISA reader (Tecan Infinite F50, Männedorf, Switzerland).

#### 2.2.4. Calculation of ELISA Assay Results

A standard curve was prepared in CombiStats Software (version 7.0, European Directorate for the Quality of Medicines & HealthCare, Strasbourg, France) using the five-parameter logistic model based on the absorbance results for the reference serum standard. The absorbance values for the tested sera were calculated into the quantity of antibody (µg/mL) in relation to the standard. The final results were presented as geometric mean antibody titers (GMT) for each tested vaccine. 

#### 2.2.5. Acceptance Criteria

The Limit of Quantification (LOQ) of the test was validated; it was 0.08 µg/mL. LOQ was estimated based on the mean absorbance value of the negative control wells plus six standard deviations (SD). All results above the LOQ are considered to be immunogenic. Samples for which the absorbance values are in the range of the absorbance values of the negative controls were recognized as non-responders. The average absorbance value obtained from the duplicate negative control wells must be less than or equal to 0.1. The slope factor of the standard curve must not be less than 0.98.

### 2.3. Statistical Analysis

Descriptive statistics for the lower confidence limit of tetanus potency and determination of anti-TT IgG antibody were presented as mean and SD, median, and range. Distribution normality testing was performed using the Shapiro–Wilk test. The overall significance of differences in results between groups was assessed (due to the non-normal distribution of results) using the Kruskal–Wallis rank-sum test. Paired comparisons were obtained using the Mann–Whitney test with Bonferroni multiple comparison corrections. Computations were performed using the R 4.2.1 statistical package (R Core Team (2022). R: A language and environment for statistical computing. R Foundation for Statistical Computing, Vienna, Austria. URL https://www.R-project.org/25.05.2023 (accessed on 20 June 2023).

## 3. Results

The study presents the results of potency assays carried out over the last five years in our OMCL (Official Medicine Control Laboratory). All the results presented met the requirements for a valid assay. The lot numbers (batches) of the tested vaccines have been coded. In accordance with the rules of good laboratory practice and the requirements of ISO 17025, Shewhart’s control charts were used to assess the consistency of the production process of the T, DT, and DTwP vaccines based on tetanus immunogenicity. 

A control chart consists of points representing a statistical analysis like mean—the center line of the control chart, which is drawn at the average value of all the results taken at different times, and standard deviation—the lines lying above and below the mean, which are the upper and lower control limits that are three variances of the mean.

The results out of the upper and lower control limits indicate deviations in the production process. 

Using Shewart’s control cards, the consistency of the technological process of tetanus component activity in the T, DT, and DTwP vaccines was assessed. The obtained results showed the consistency of production control of Polish tetanus-containing vaccines.

We observed that, for the tetanus vaccine (trade name Tetana), the mean of 22 results of the lower confidence limit of tetanus potency is 52.74 IU/0.5 mL, and the standard deviation is 11.64 (Figure 1). For the diphtheria–tetanus vaccine (trade name DT), the mean of seven results of the lower confidence limit of tetanus potency is 46.41 IU/0.5 mL, and the standard deviation is 8.74 (Figure 2). For the diphtheria–tetanus–pertussis vaccine (trade name DTP), the mean of 11 results of the lower confidence limit of tetanus potency is 231.16 IU/0.5 mL, and the standard deviation is 60.57 (Figure 3). It was noticed that the lower confidence limit of tetanus potency in the DTwP vaccine was more than four times higher than the lower confidence limit of tetanus potency in the DT and T vaccines, despite the same amount of tetanus antigen in the vaccine formulations.

A statistical analysis showing significant differences between the groups is presented below. Descriptive statistics of results are presented in Table 2 and Figure 4. A significant overall difference between groups was found (*p* < 0.001). Pairwise comparisons between groups (after applying Bonferroni correction) revealed statistically significant differences between T and DTP and DT and DTP groups. No statistically significant difference between T and DT groups was found.

## 4. Discussion

Vaccination plays a huge role in the eradication and prevention of infectious diseases. Vaccines can prevent prolonged infections by stimulating the immune response [4]. In this study, an adjuvant effect of the whole-cell pertussis component on the immune response against tetanus was observed. In the murine model, it was noticed that the potency of the tetanus component in the DTwP vaccine was significantly higher than tetanus potency in DT and T vaccines, despite the same amount of tetanus antigen Lf in the tested vaccine formulations. Numerous studies have described the effectiveness of whole-cell pertussis as an adjuvant when administered in combination with various antigens [9,10,11]. The adjuvant impact of the *B. pertussis* cells on antibodies against diphtheria toxoid production was confirmed for the first time in the 1940s [12]. Oliveira et al. investigated the adjuvant effect of the whole-cell pertussis component on the pneumococcal surface protein A (PspA) vaccine. The intranasal immunization of mice with the experimental combination PspA-wP vaccine provided better protection against a lethal challenge with the *Streptoccocus pneumoniae* reference strain than the PspA vaccine given alone. Additionally, the high levels of anti-pneumococcal antibodies led to cross-protection against various pneumococcal serotypes. Similar levels of protection were observed when the aforementioned vaccines were administered subcutaneously. The mice were protected from a nasal challenge with two different pneumococcal strains [11]. The immunization of mice with the wP vaccine in combination with influenza antigens has been shown to induce humoral responses against influenza and antigen-specific IL-2 and IFN-γ production [10,13,14]. Studies using the murine model have demonstrated the adjuvant impact of the wP vaccine on the subunit influenza vaccine. The influenza antibody induction level is 250 times higher after immunization with the influenza vaccine administered together with the whole-cell pertussis component compared with a vaccine dissolved only in saline. Protection against a challenge virus infection was also provided [14]. In children in Slovakia, vaccination both with the DTwP and hepatitis B (HBV) vaccines together resulted in a higher humoral response to HBV compared with children who only received the HBV vaccine [15]. The adjuvant effect of the DTwP vaccine on the response against *Haemophilus influenzae* type b (Hib) was observed in studies conducted by Nicol et al. Children given 1/10 of the dose of the Hib conjugate vaccine diluted in the DTwP vaccine had similar antibody levels compared to children vaccinated with the full dose of the Hib conjugate vaccine only [16]. It has been shown that the co-administration of the seven-valent pneumococcal vaccine together with the DTwP vaccine stimulates a higher immune response to pneumococcal polysaccharides in French babies [17]. Studies with the 11-valent pneumococcal vaccine have shown a weaker immune response to pneumococcal polysaccharides when the vaccine was administered with the DTaP (acellular pertussis component) inactivated polio (IPV) and Hib vaccines. The effect of a reduced immune response against pneumococcal polysaccharides was not observed when the whole-cell pertussis component was used [18].

Our paper presents the immune response against tetanus in mice vaccinated with monovalent and combined tetanus toxoid vaccines. The challenge test in mice determines the potency of a vaccine based on its ability to induce both cellular and humoral immune responses. In our studies, we did not observe higher levels of anti-TT antibodies in tested mice after DTP vaccination compared to the T and DT vaccines. Our results suggest that the significantly higher potency of the DTP vaccine in stimulating the immune response against tetanus is not related to the stimulation of the humoral immune response. In this paper, we suggest that the potential way of the pertussis component adjuvant activity is to stimulate a cellular immune response. CTL responses are usually induced by cellular vaccines, including the whole-cell pertussis vaccine (wP) [19,20,21,22]. Following intranasal immunization of volunteers with the wP vaccine, specific T-cell responses and IgA responses against pertussis antigens were induced [20]. Understanding the mechanism of action of the adjuvant effect of the whole-cell pertussis component is still being discussed. According to many studies [23,24,25], lipopolysaccharide (LPS) is responsible for the adjuvant effect on the immune system. LPS is the main molecule located on the outer membrane of Gram-negative bacteria. LPS consists of O-specific polysaccharide, core oligosaccharide, and lipid A [24]. Lipid A shows the greatest biological activity, including adjuvant and toxicity [23]. The adjuvant effect of LPS was first shown in 1956 [23]. LPS stimulates the production of growth factors and inflammatory and proinflammatory mediators such as IL-6, IL-12, IFN-γ, and TNF-α. LPS also increases the Th1/Th2 ratio and enhances antigen uptake, processing, and presentation [23]. LPS exhibits strong biological activity, including anti-tumor activity, effects on pyrogenicity, and activation of complement, macrophages, and granulocytes [25]. The adjuvant properties of LPS were shown for humoral [23] and cell-mediated immunity [26]. It has already been shown that monophosphoryl lipid A isolated from the *B. pertussis* strain is a good adjuvant for the influenza vaccine [23]. The authors of the publication [14] also suspect that LPS present in *B. pertussis* cells is responsible for the adjuvant effect of the whole-cell pertussis component [14]. LPS stimulates immune responses through the TLR2 and TLR4 receptors [23], which was confirmed in numerous studies [11,19,27] where TLR4 responses were important for wP-induced Th1 and Th17 responses. Other studies have already demonstrated that LPS derived from *B. pertussis* isolates induces TLR4 responses but cannot stimulate TLR2 responses [28,29]. In vitro studies on human cells have shown that purified *B. pertussis* LPS induces Th2-dependent immune responses via TLR4 [28]. Arora et al. [30] showed that the vaccination of rats with tetanus toxoid (TT) together with LPS of Pseudomonas aeruginosa significantly increases the level of anti-TT IgG compared with the administration of TT alone. They also observed that the adjuvant effect of LPS on the humoral response against TT occurs through the production of TNF-α. The adjuvant impact of LPS on protein antigens involves T-cell-independent B cell proliferation [31], but the adjuvant effect of LPS on polysaccharide antigens has been associated with the regulation of T lymphocytes, particularly T suppressor cells [32]. Tetanus toxoid is a thymus-dependent antigen and requires T-cell cooperation with B cells to stimulate specific anti-TT antibodies [23]. According to the results of Mohammadi’s study, unconjugated LPS better enhanced the activity of TT than the conjugated product (TT-LPS). TT-LPS could not produce high antibody titers in mice [33]. The study conducted by Kariminia et al. [34] showed that LPS from Brucella induces IL-10 and IL-12 production and the priming of IFN-γ in human peripheral blood mononuclear cell culture. Interleukin 12 and IFN-γ are responsible for the differentiation of T lymphocytes to Th1 cells. 

In contrast, in Oliveira’s study described above, two experimental vaccines were used—the PspA-wP vaccine and the PspA-wP low vaccine (which contained a low level of *B. pertussis* LPS). Both vaccines induced comparable protective levels of mucosal anti-pneumococcal antibodies, indicating no effect of *B. pertussis* LPS on the adjuvant properties of the whole-cell pertussis component [11]. The authors also showed that in the absence of TLR4, the immune response is sufficient to induce protection against a pneumococcal nasal challenge. They conclude that other pertussis virulence factors, such as pertussis toxin and adenylate cyclase toxin, may also act as adjuvants for a pneumococcal immune response [11]. Other studies [35,36,37] have proven the adjuvant effect of pertussis toxin. Mice immunized with the inactivated influenza vaccine formulated with the purified B oligomer of pertussis toxin stimulated the production of specific anti-haemagglutinin antibodies and mucosal IgA antibody. Additionally, mice were protected against the challenge with the influenza virus [37]. 

Our previous study [38] has shown no pertussis toxin in the Polish DTwP vaccine. In mice immunized with the national DTwP vaccine, no activation of antibodies against the pertussis toxin was observed [38]. Since 1978, the Polish DTwP vaccine has been manufactured using three vaccine seeds of *B. pertussis* strains isolated in the 1960s. *B. pertussis* production strains were cultured on an H3 liquid medium and were inactivated using chemical (with 0.12% formaldehyde) and thermal (incubation at 25 °C for 24 h) methods. The suspensions of inactivated *B. pertussis* strains have been stored for eight months at 4 °C for detoxification. This is the time required for the toxic antigens to penetrate into the supernatant. Subsequently, the detoxicated strains were combined with purified tetanus and diphtheria toxoids adsorbed on alum. 

The composition of most wP vaccines is not clearly defined and depends on the production technology. Different content of all *B. pertussis* antigens, including pertussis toxin, dermonecrotic toxin, adenylate cyclase toxin, tracheal cytotoxin, filamentous haemagglutinin, pertactin, agglutinogens, lipopolysaccharide, tracheal colonization factor, type III secretion system, and Bordetella resistance to killing, was observed [39,40]. Probably, all known antigens listed above are responsible for the adjuvant impact of the wP vaccine but also contribute to adverse events observed after wP vaccination [39,40,41]. Perhaps, some toxins which are detected as residues retain their immunogenicity and reactogenicity. The differences in the reactogenicity of many wP vaccines in different countries are due to differences in the number of virulence factors between vaccine preparations, resulting from differences in the inactivation and detoxification process during the production as well as strains used for vaccine production [42,43]. The adjuvant effect of wP vaccines also influenced other, so far undescribed, *B. pertussis* antigens. The wP vaccines contain, apart from commonly known antigens, many other molecules with immunogenic properties. In 2009, Altindis et al. [44] identified 25 *B. pertussis* antigens, including 21 new immunogenic proteins that may determine their high activity. Research conducted by Koenig et al. [45] aligns with the above findings. The authors proved that the adjuvant effect of *B. pertussis* is mediated by killed whole *B. pertussis* cells [45].

The development of adjuvants is an important issue linked to understanding their methods of action. Despite their long history, adjuvants have long been used without a clear understanding of how they work [5,46]. This lack of knowledge has inhibited vaccine development in terms of inventing more effective and safer adjuvants. Currently used vaccines induce mainly a strong humoral response while weakly stimulating the cellular T-cell response. Alum used in the study vaccines promotes antibody responses. The mechanism of adjuvant effects mediated by alum includes the antigen depot effect. Presented antigens are slowly released from the vaccination sites and then are uptaken by antigen-presenting cells (APCs) [47,48]. Antigens remaining at the injection site induce local inflammation involving the production of interleukins IL-1b and IL-18, which strongly polarizes antibody responses and Th2 immunity [49,50,51]. Th2-type immune responses are not effective in inducing cell-mediated immunity and IgA antibody-dependent mucosa responses [52]. Many studies are focused on describing new adjuvants that also induce cellular immunity [53] because balanced humoral and CTL immunity is important for protection against most pathogens [54].

In the future, it would be worth examining the cellular immune response as the most probable mechanism of action of the adjuvant effect of whole-cell pertussis on tetanus toxoid.

## Figures and Tables

**Figure 1 biomedicines-11-01795-f001:**
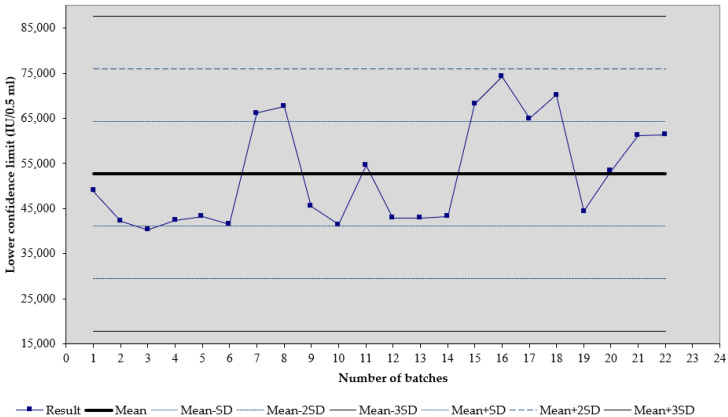
Shewart’s control chart of the lower confidence limit of tetanus potency results in T vaccines. Results are shown as mean (IU/0.5 mL) plus SD.

**Figure 2 biomedicines-11-01795-f002:**
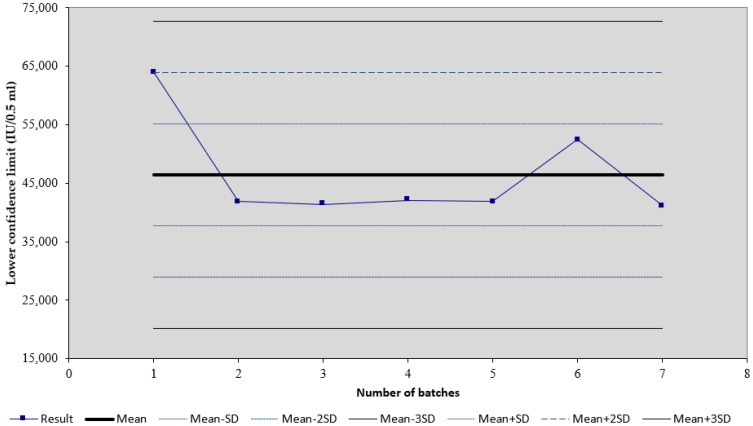
Shewart’s control chart of the lower confidence limit of tetanus potency results in DT vaccines. Results are shown as mean (IU/0.5 mL) plus SD.

**Figure 3 biomedicines-11-01795-f003:**
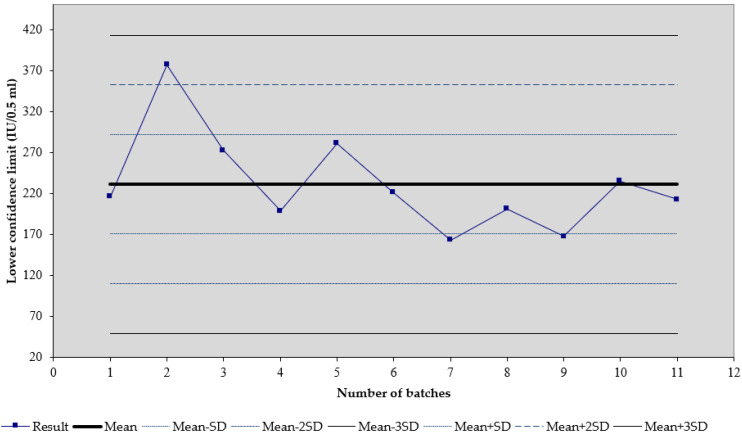
Shewart’s control chart of the lower confidence limit of tetanus potency results in DTP vaccines. Results are shown as mean (IU/0.5 mL) plus SD.

**Figure 4 biomedicines-11-01795-f004:**
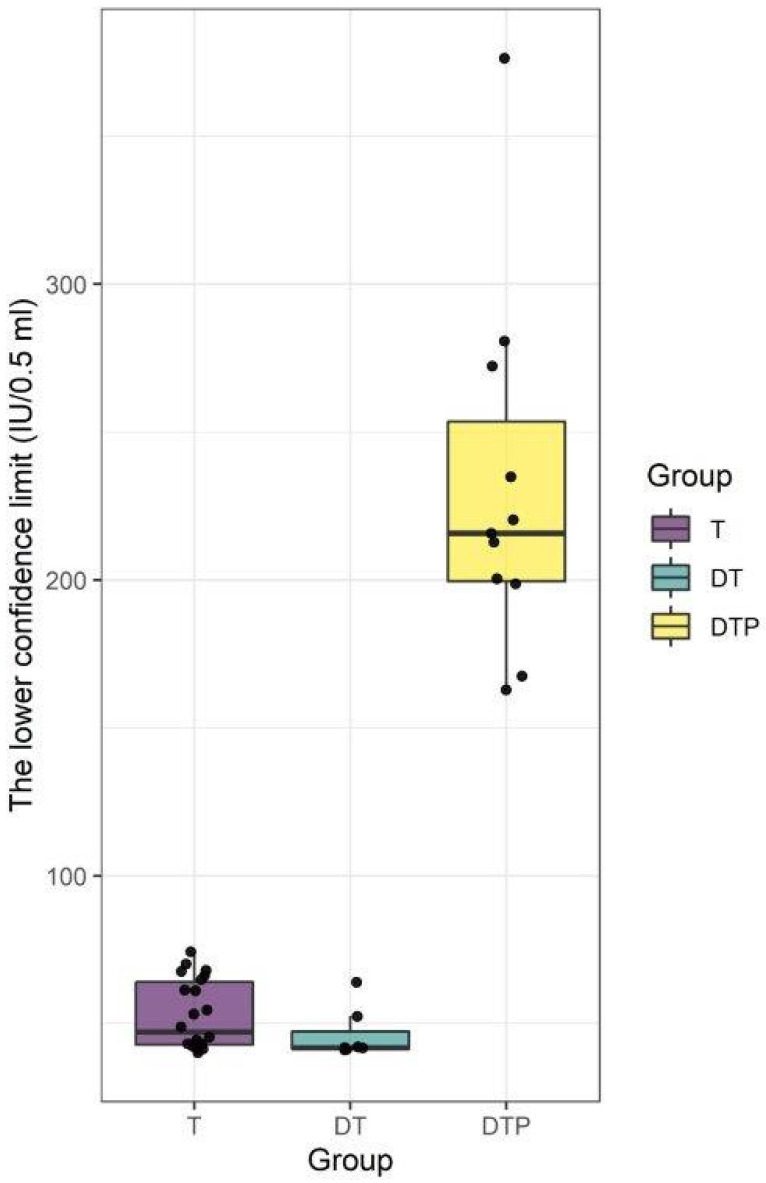
Statistical analysis. Challenge test in mice. Results (dots) are shown as the lower confidence limit (IU/0.5 mL) for each tested vaccine. The rectangles represent the means and ranges of results for each tested group. Statistically significant differences between T and DTP and DT and DTP groups were found (*p* < 0.001). No statistically significant difference between T and DT groups was found. In our study, the antibody response after immunization with monovalent and combined T vaccines was evaluated. Using the ELISA method, we demonstrated that IgG anti-TT were induced by all vaccines used in the study at the same levels—for the T vaccine, GMT of the anti-TT antibodies were at the level 18.82 µg/mL, for the DT vaccine, GMT of the anti-TT antibodies were at the level 18.75 µg/mL, and for the DTP vaccine, GMT of the anti-TT antibodies were at the level 18.62 µg/mL (Figure 5). All vaccinated mice responded to immunization against tetanus. No statistically significant differences were observed (*p* > 0.9). A statistical analysis showing no significant differences between the groups is presented below. Descriptive statistics of results are presented in Table 3 and Figure 6.

**Figure 5 biomedicines-11-01795-f005:**
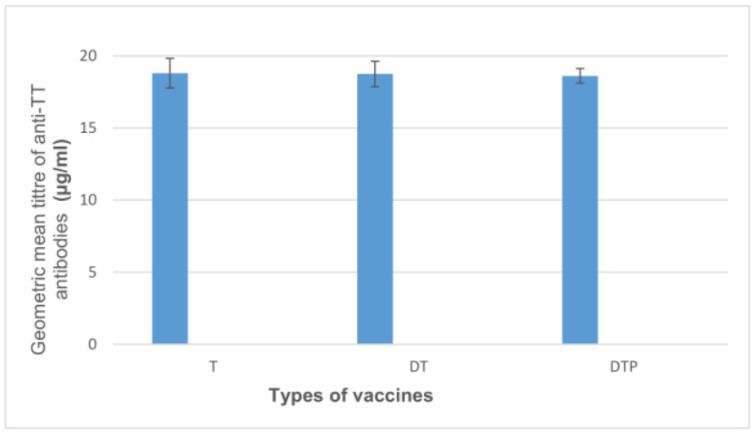
The levels of anti-TT antibody after immunization with monovalent and combined T vaccines.

**Figure 6 biomedicines-11-01795-f006:**
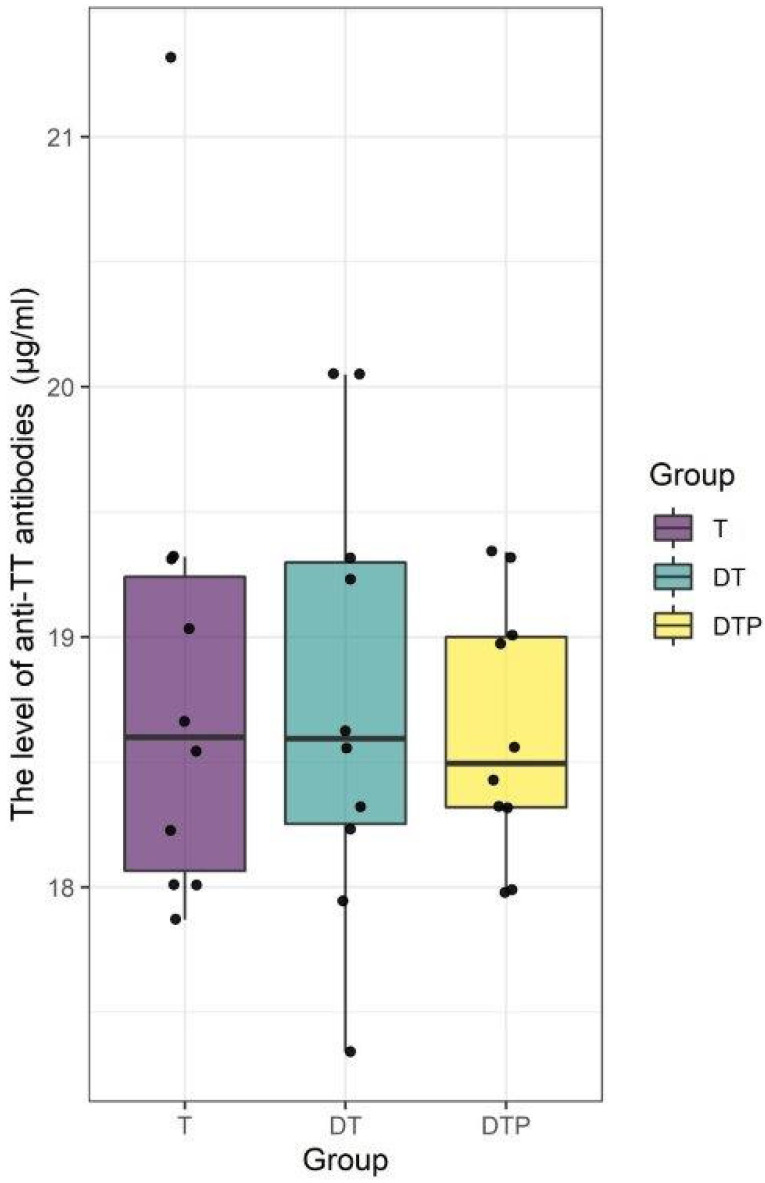
Statistical analysis. Assessment of anti-TT IgG levels in mouse sera. Results (dots) are shown as the anti-TT level (µg/mL) for each vaccinated mouse. The rectangles represent the means and ranges of results for each tested group. No statistically significant differences between T, DT, and DTP groups were found (*p* > 0.9).

**Table 1 biomedicines-11-01795-t001:** Vaccines used in the study.

Type of Vaccine	Composition of a Single Human Dose (0.5 mL)
T vaccine	• Tetanus toxoid 10 Lf adsorbed on 0.5 mg hydrated aluminum hydroxide
DT vaccine	• Tetanus toxoid 10 Lf adsorbed on 0.45 mg hydrated aluminum hydroxide • Diphtheria toxoid 20 Lf adsorbed on 0.45 mg hydrated aluminum hydroxide
DTP vaccine	• Tetanus toxoid 10 Lf adsorbed on 0.45 mg hydrated aluminum hydroxide • Diphtheria toxoid 20 Lf adsorbed on 0.45 mg hydrated aluminum hydroxide • Suspension of inactivated *Bordetella pertussis* strains—15 IOU

**Table 2 biomedicines-11-01795-t002:** Statistical analysis. Challenge test in mice.

	Tested Vaccines			*p*-Value ^1^
	T, N = 22	DT, N = 7	DTP, N = 11	
Result				<0.001
N	22	7	11	
Mean (SD)	52.7 (11.6)	46.4 (8.7)	231.2 (60.6)	
Median	47.2	41.9	215.8	
Range	40.2–74.3	41.2–64.0	162.9–376.4	

^1^ Kruskal–Wallis rank sum test.

**Table 3 biomedicines-11-01795-t003:** Statistical analysis. Assessment of anti-TT IgG levels in mouse sera.

	Tested Vaccines			*p*-Value ^1^
	T, N = 10	DT, N = 10	DTP, N = 10	
Result				>0.9
N	10	10	10	
Mean (SD)	18.8 (1.0)	18.8 (0.9)	18.6 (0.5)	
Median	18.6	18.6	18.5	
Range	17.9–21.3	17.3–20.0	18.0–19.3	

^1^ Kruskal–Wallis rank sum test.

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
