# Peer review of "Adjuvant Effect of Whole-Cell Pertussis Component on Tetanus Toxoid Potency in Murine Model"

_biomedicines, 2023, doi:10.3390/biomedicines11071795_

Round 1

Reviewer 1 Report (Previous Reviewer 1)

In their revised manuscript, Prygiel et al. show data on the potency of different tetanus toxoid vaccine formulations and have now included data on the antibody responses after the different vaccinations, which I appreciate.

Please find a way to show data only once, it is not necessary to show the same data in 2 or 3 graphs and tables. Also information on statistical significance can be incorporated into one figure.

The figure legends are incomplete, at present there are only figure legend titles.

Author Response

Dear Editors,

Thank you for your valuable comments.

Editor 1

In their revised manuscript, Prygiel et al. show data on the potency of different tetanus toxoid vaccine formulations and have now included data on the antibody responses after the different vaccinations, which I appreciate.

Please find a way to show data only once, it is not necessary to show the same data in 2 or 3 graphs and tables. Also information on statistical significance can be incorporated into one figure.

The figure legends are incomplete, at present there are only figure legend titles.

Figures showing the estimated potency of vaccines have been removed (they do not introduce new information). The display of repeated data may cause difficulties in understanding for the addressees of the manuscript. It will be difficult to show the statistics from both experiments in one figure. The statistics are made individually and the program draws a separate graph for each calculation.

Figure legends have been introduced as suggested.

Marta Prygiel

Reviewer 2 Report (Previous Reviewer 2)

The manuscript is much improved in light of reviewers comments. The manuscript can now be accepted for publication. 

Minor English check needed. 

Author Response

Dear Editors,

Thank you for your valuable comments.

Editor 2

The manuscript is much improved in light of reviewers comments. The manuscript can now be accepted for publication. Minor English check needed. 

Minor linguistic corrections have been made.

Marta Prygiel

Reviewer 3 Report (Previous Reviewer 3)

The paper presentation needs to be fixed. It is hard to understand as current. The main issue is the figure presentation.

 The Figure order is unusual. It was demonstrated as Figure 1a, 2a, 3a and 1b, 2b, 3b, then followed by Figure 4, 5 and 6.  Please fix it. The presentation of the figures is a little bit mass Figure 4, 5 and 6 are too big and the figures are blurry. There is no label for the y axis of Figure 4-6. Please use “.” for decimal instead of “,”, wherever it is applicable.

Some examples :

Line 100 pH 7,4  pH7.4

Line 168 0,08 µg/ml  0.08 µg/m

Line 175 “0.1 The”  0.1. The

Figure 1a, 2a, 3a, y axis label and number  

The y axis unit number

Line 241-245 the antibody levels

Line 216: Is that “Figure 4” instead of Figure 3?

same as above

Author Response

Dear Editors, 

Thank you for your valuable comments,

Editor 3

The paper presentation needs to be fixed. It is hard to understand as current. The main issue is the figure presentation.

 The Figure order is unusual. It was demonstrated as Figure 1a, 2a, 3a and 1b, 2b, 3b, then followed by Figure 4, 5 and 6.  Please fix it.

Figures 1a and 1b, 2 a and 2b, 3a and 3b represent the same data shown in two ways, therefore figures showing the estimated potency of vaccines (1b, 2b, 3b) have been removed (they do not introduce new information).

The presentation of the figures is a little bit mass Figure 4, 5 and 6 are too big and the figures are blurry. There is no label for the y axis of Figure 4-6.

The size of the figures has been reduced. Description under statistical analysis figures has been added, to better understand what is shown in the figures. The labels for the y axis of Figure 4-6 have been added.

Please use “.” for decimal instead of “,”, wherever it is applicable. – corrected.

Some examples :

Line 100 pH 7,4  pH7.4 corrected

Line 168 0,08 µg/ml  0.08 µg/m corrected

Line 175 “0.1 The”  0.1. The corrected

Figure 1a, 2a, 3a, y axis label and number  corrected

The y axis unit number corrected

Line 241-245 the antibody levels corrected

Line 216: Is that “Figure 4” instead of Figure 3? corrected

Best,

Marta Prygiel

This manuscript is a resubmission of an earlier submission. The following is a list of the peer review reports and author responses from that submission.

Round 1

Reviewer 1 Report

In their current manuscript, Prygiel et al. present data from tetanus immunizations of mice where three types of vaccines are compared. While the data is not surprising, it is of some general interest, but the presentation of the data should be improved and supplemented with data on antibody responses. These are my specific suggestions:

Experimental:

The authors only show data on the protection elicited by the different vaccines, this should be complemented by antibody data from the immunized mice before the toxin challenge. Are there significant differences?

On the statistical side, the authors currently state in the discussion that there was a significantly higher tetanus potency in the DTwP vaccine compared to the other two. They should provide information on the test performed and the p values obtained, maybe as a table.

Data presentation:

The authors basically show the same data twice, since Fig. 1 shows the low confidence levels of the data shown in Fig. 4 and so on. This should be amended, maybe by making Fig. 4 into Fig. 1A and Fig. 1 into Fig. 1B.

In the current Fig. 1, one data point seems to have been skipped to the right -> there is no value for batch 4 but there is a batch 23 which is not there in Fig. 4 and according to the statement that 22 batches were tested.

Figures 1-3: why are there pink data points on the black line that indicates the mean of all the other data points, surely there is only one mean value and the line is enough to indicate that.

Introduction:

Please provide some background and rationale for the comparison you are making, i.e. checking for an adjuvant effect. Also reference work from others here that have performed similar studies.

Discussion:

Since the authors fail to mention an adjuvant effect in the whole results section, the first very long section on adjuvants seems disconnected. Please restructure.

The authors state that first experiments showing an adjuvant effect of B. pertussis on diphteria toxoid was confirmed in the 1950s but cite a paper from 1962. 

line 276: the authors state incubation at 25°C as a thermal inactivation method, that seems a very ambient temperature, please check if correct.

The results presented in this manuscript are not surprising and not that detailed, and I don't feel that the main conclusion proposed at the end of the discussion (lines 302-306) is justified.

Author Response

Dear Editor,

Thank you for your valuable comments. They have been introduced as suggested as far as possible.

Experimental:

The authors only show data on the protection elicited by the different vaccines, this should be complemented by antibody data from the immunized mice before the toxin challenge. Are there significant differences?

We have not conducted such studies at this time and it is not possible to add antibody data at this stage of revision of the manuscript. It is a very valuable clue for expanding research. The statement that it is worth investigating was added in the conclusion of the discussion of the article.

On the statistical side, the authors currently state in the discussion that there was a significantly higher tetanus potency in the DTwP vaccine compared to the other two. They should provide information on the test performed and the p values obtained, maybe as a table.

The statistics have been made.

Data presentation:

The authors basically show the same data twice, since Fig. 1 shows the low confidence levels of the data shown in Fig. 4 and so on. This should be amended, maybe by making Fig. 4 into Fig. 1A and Fig. 1 into Fig. 1B.

The figures has been corrected.

In the current Fig. 1, one data point seems to have been skipped to the right -> there is no value for batch 4 but there is a batch 23 which is not there in Fig. 4 and according to the statement that 22 batches were tested.

The figure has been corrected.

Figures 1-3: why are there pink data points on the black line that indicates the mean of all the other data points, surely there is only one mean value and the line is enough to indicate that.

The figures has been corrected.

Introduction:

Please provide some background and rationale for the comparison you are making, i.e. checking for an adjuvant effect. Also reference work from others here that have performed similar studies.

The introduction has been edited to make it more readable.

Discussion:

Since the authors fail to mention an adjuvant effect in the whole results section, the first very long section on adjuvants seems disconnected. Please restructure.

Some things from the discussion have been moved to the introduction.

The authors state that first experiments showing an adjuvant effect of B. pertussis on diphteria toxoid was confirmed in the 1950s but cite a paper from 1962. 

Farthing and colleagues in a 1962 article mentions that Greenberg and Fleming in 1947 and 1948 observed the adjuvant effect of a pertussis component on diphtheria toxoid. The 1950s have been changed to the 1940s.  line 276: the authors state incubation at 25°C as a thermal inactivation method, that seems a very ambient temperature, please check if correct.

I checked this information in the manufacturer's production and control protocol. Biomed (manufacturer) use a temperature of 25°C and call it thermal inactivation. Should the word “thermal” be removed?

The results presented in this manuscript are not surprising and not that detailed, and I don't feel that the main conclusion proposed at the end of the discussion (lines 302-306) is justified.

The main conclusion has been removed. In conclusion, we proposed additional research that would be worth doing in the future.

With regards,

Marta Prygiel

Reviewer 2 Report

Thank you for referring me this paper for review. 

The paper is very short and important explanations are missing throughout the paper. My suggestion will be to improve overall text of the paper so the readers can clear follow the study objectives. 

For example, the introduction is very short and difficult to understand what the authors are looking to do. 

The methods sections also need to be redmodelled for reproducibility.

The figures quality can be improved. 

I am not seeing the choice of adjuvent used?

Is a single adujvent is employed or different?

Author Response

Dear Editor,

Thank you for your valuable comments. They have been introduced as suggested as far as possible.

The paper is very short and important explanations are missing throughout the paper. My suggestion will be to improve overall text of the paper so the readers can clear follow the study objectives. 

For example, the introduction is very short and difficult to understand what the authors are looking to do. 

The introduction and discussion have been edited to make it more readable.

The methods sections also need to be redmodelled for reproducibility.

The materials and methods section has been described as described in European Pharmacopeia.

The figures quality can be improved. 

The figures has been changed.

I am not seeing the choice of adjuvent used?

Is a single adujvent is employed or different?

The vaccines used in this study contain aluminium hydroxide (alum) as an adjuvant.

This has been described in the introduction as well as in the Table 1. Vaccines used in the study.

With regards,

Marta Prygiel

Reviewer 3 Report

The paper tested the potency of 40 batches of vaccines, 22 tetanus (T) vaccines, 11 diphtheria-tetanus-whole-cell pertussis (DTP), 7 diphtheria-tetanus (DT) vaccines against the reference vaccine, AnaTe9, in Swiss mice (n=10/group). The results demonstrated that the potency of the tetanus component in the DTP vaccine was significantly greater than that in DT and T vaccines. However, the paper only summarized the observed results, did not investigate any possible mechanisms of the different potency of the tetanus component in the DTP.

 Major comments:

If possible, please investigate why the tetanus component is more potent in the DTP than it is used alone in the T vaccines or combined with diphtheria in the DP vaccines.

Minor comments:

1.       International Units (IU): please keep consistency throughout the context.

2.       Please check the reference formatting, i.e. ref 6-10, 12-16, 26-32, 36-38, 49-51.

3.       Please check the reference journal abbreviation format, and keep the abbreviation format consistent.

4.       Ref 36: “BMC 2008 Immunol” should be “BMC Immunol 2008”

5.       Ref43: “Infec.t” should be”infect.”.

Author Response

Dear Editor,

Thank you for your valuable comments. They have been introduced as suggested as far as possible.

Major comments:

If possible, please investigate why the tetanus component is more potent in the DTP than it is used alone in the T vaccines or combined with diphtheria in the DP vaccines.

We have not conducted such studies at this time mechanism of action of whole cell pertussis component at this stage of revision of the manuscript. It is a very valuable clue for expanding research. The statement that it is worth investigating was added in the conclusion of the discussion of the article.

Minor comments:

  1. International Units (IU): please keep consistency throughout the context.- corrected
  2. Please check the reference formatting, i.e. ref 6-10, 12-16, 26-32, 36-38, 49-51.- corrected
  3. Please check the reference journal abbreviation format, and keep the abbreviation format consistent. -checked
  4. 4.Ref 36: “BMC2008 Immunol” should be “BMC Immunol 2008” - corrected
  5. Ref43: “Infec.t” should be”infect.”. - corrected

With regards,

Marta Prygiel

Round 2

Reviewer 1 Report

In the revised version of the manuscript, the authors have made some changes to the text and added a statistical analysis, whereas the figures have not been changed, contrary to the authors' statement, and no additional data has been provided. In the current form, I do not support acceptance of the manuscript. I believe that it is important to show antibody data, and suggest that the authors perform such analyses in the next possible instance. Since they do the protection tests as part of the quality control for vaccine batch release, they will probably have the opportunity in the near future. That would significantly strengthen the value of the manuscript and provide mechanistic insight.

Reviewer 2 Report

Accept